# Numbers and types of neurological emergencies in England and the influence of socioeconomic deprivation: a retrospective analysis of hospital episode statistics data

Michael Jackson,[1] Marta Szczepaniak,[1] Jasmine Wall,[2,3] Mark Maskery,[2,3] Catherine Mummery,[4] Paul Morrish,[5] Adrian Williams,[6,7] Joanne Knight [ID],[2] Hedley C A Emsley [ID] [2,3]

For numbered affiliations see end of article.

**Correspondence to**
Hedley C A Emsley;
hedley.emsley@lancaster.ac.uk

## ABSTRACT

**Objectives** In this first large-scale analysis of neurological emergency admissions in England, we determine the number and types of emergency admissions with neurological emergency diagnostic codes, how many are under the care of a neurologist or neurosurgeon and how such admissions vary by levels of deprivation.

**Design** Retrospective empirical research employing a derived list of neurological emergency diagnostic codes

**Setting** This study used the Hospital Episode Statistics data set for the financial year 2019/2020 based on 17 million in-year inpatient admissions in England including 6.5 million (100%) emergency admissions with any diagnosis codes.

**Results** There were 1.4 million (21.2%) emergency inpatient admissions with a mention of any neurological code, approx. 248 455 (3.8%) with mention of a specific neurological emergency code from the derived list, and 72 485 (1.1%) included such a code as the primary reason for admission. The highest number of in-year admissions for adults was for epilepsy (145 995), with epilepsy as the primary diagnostic code in 15 945 (10.9%). Acute nerve root/spinal cord syndrome (41 215), head injury (29 235) and subarachnoid haemorrhage (18 505) accounted for the next three highest number of admissions. 3230 (1.4%) in-year emergency hospital admissions with mention of a neurological emergency code were under the care of a neurologist or neurosurgeon, with only 1315 (0.9%) admissions with mention of an epilepsy code under a neurologist. There was significant variation for epilepsy and functional neurological disorders (FNDs) in particular by Index of Multiple Deprivation decile. The association between deprivation and epilepsy and FND was significant with p-values of 2.5e-6 and 1.5e-8, respectively.

**Conclusions** This study has identified important findings in relation to the burden of neurological emergency admissions but further work is needed, with greater clinical engagement in diagnostic coding, to better understand the implications for workforce and changes to service delivery needing to be implemented.

## STRENGTHS AND LIMITATIONS OF THIS STUDY

⇒ First analysis of its type arising from an academic partnership with the Office for Health Improvement and Disparities, utilising Hospital Episode Statistics data.
⇒ Large-scale analysis based on 6.5 million emergency admissions increases robustness of findings.
⇒ Data used were collected for the purposes of healthcare delivery rather than research.
⇒ Retrospective nature of study limits interpretation of some findings such as the nature of suspected cauda equina presentations.
⇒ Inclusion of emergency admissions with mention of neurological condition rather than solely neurological emergencies per se on account of coding accuracy concerns.

## INTRODUCTION

Medical emergency admissions continue to rise across the UK. They are more than twice as likely among the most deprived population decile by comparison with mean deprivation, reinforcing the point that challenges are social as well as medical.[1 2] Many emergency admissions to hospital are considered to be avoidable and patients often stay in hospital longer than is necessary.[3] Emergency admissions are also disruptive of elective care delivery, prolonging waiting times. Hospital Episodes Statistics (HES) data for 2016/2017 suggest that approximately 18% of medical emergency admissions included a mention of neurological conditions including stroke and dementia, totalling 1.09 million, a rise of 21% over the previous 5 years.[4] Excess mortality has also been reported among patients with epilepsy.[5] At a time when costs are rising and resources are increasingly under pressure, it is all the more important to have a better

understanding of neurological emergency admissions and underlying factors such as social deprivation.

There has been little by way of systematic evaluation of data pertaining to neurological emergency admissions. Previous work has used a much broader definition, counting neurological conditions present in patients admitted as an emergency, including stroke and dementia, rather than neurological emergencies per se. The present work has by necessity involved the creation of a list defining neurological emergencies. Much remains unknown about neurological emergency admissions, including their total number, the distribution of neurological conditions and the effect of socioeconomic deprivation. There have been some studies investigating the influence of comorbidities and race in the USA.[6][7] This study represents the first large-scale analysis of neurological emergency admissions in England, and the first OHID-academic partnership in neurology. For the first time, the data provide some insights into the number and types of emergency admissions with neurological emergency codes, the number of such admissions under the care of a neurologist and how such admissions vary on the basis of factors such as levels of deprivation.

## METHODS
### Defining neurological emergencies
For the purposes of this work, it was necessary to create a list of conditions typically regarded as neurological emergencies, and to map these to the relevant ICD-10 codes. An initial set of neurological conditions was proposed by AW as the basis for the study, comprising causes of acute paralysis such as Guillain-Barré syndrome, myasthenic crisis, encephalitis, spinal cord compression and status epilepticus. The ICD-10 codes most closely matching these 'index' entities were compiled into a list. Other ICD-10 codes in close hierarchical proximity to the 'index' ICD-10 codes were added from a previously published Public Health England neurology code collection,[8] which comprised 581 4-digit ICD-10 codes for 19 condition groups. The 'long list' of candidate neurological emergency codes, consisting of 136 ICD-10 codes, was reviewed iteratively by the project team and clinicians working alongside, and refined into a short list of 96 4-digit ICD-10 codes (online supplemental appendix 1) for use in the work reported here, subsequently referred to as the 'derived list' (of neurological emergency codes). The selection process used clinical judgement to decide which conditions or groups of conditions to include or exclude, with reference to entities classically considered to be neurological emergencies. Of particular note, stroke, despite being a classical neurological emergency, was excluded from this exercise given that stroke is already the subject of the existing national audit system (SSNAP). Codes for conditions presenting with secondary headache have been included; codes for primary headache disorders have not been included in the present work. Also of note was the decision to include functional neurological

### Box 1 Relevant terminology

People admitted to hospital will always have a responsible medical officer (consultant) assigned for the time under that consultant, or episode (consultant episode).

The consultant episode at the start of the hospital stay is the admission episode; the episode at the end of the hospital spell is called the discharge episode.

Hospital admissions are the start of hospital spells, which comprise one or more consultant episodes.

ICD-10—International Statistical Classification of Diseases and Related Health Problems (ICD) (10th edition) codes, or hospital diagnosis codes. ICD-10 is a legally mandated health data standard.

Conditions present during the consultant episode can be described using up to 20 corresponding ICD-10 codes. Hospital stays with multiple consultant episodes can have very differing collections of ICD-10 codes recorded on each episode.

The ICD-10 code that is recorded in the first position on a consultant episode is known as the primary diagnosis. Subsequent codes on the same episodes are considered secondary diagnoses and may represent conditions of lesser importance for this particular consultant episode or other long-term conditions/comorbidities of the patient.

Hospital episode statistics (HES) are financial year based. Stays in hospital can cross the financial year timelines, therefore the number of admissions and discharges in any one financial year may not be equal. In-year hospital admissions relates to the count of all hospital stays that start in the financial year. In-year hospital discharges relate to the count of all hospital stays that end during the financial year. HES ID is a pseudonymised unique identifier for each individual patient, which can be used to group hospital episodes and spells by individual.

disorders (FNDs) on the basis that we believed these to contribute significantly to the total number of emergency hospital admissions for neurological conditions, as well as the need for patients with FND to be referred to specialists with expertise in neurological diagnosis.[9] FNDs is a term given to neurological features that are not due to any physical neurological disorder. We have defined relevant terminology in Box 1.

### Defining study group
The hospital activities included in this study were admissions to hospital occurring during the financial year 2019/2020 in England—described as an in-year admission, where the admission was classed as an emergency (non-elective) admission to hospital and where the selected neurological emergency codes are in any diagnosis position, on any consultant episode during the hospital stay.

### Data source, data extraction and statistics
The data source used in this study is the Admitted Patient Care version of the HES data set for the financial year 2019/2020. Data extraction was undertaken by the OHID Neurology Dementia Intelligence team using Structured Query Language with further data manipulation, including descriptive statistics, using Microsoft Excel. Counts and percentages were extracted and/or calculated for various categories, CIs were estimated and data

**Table 1** Inpatient hospital activity by admission type; all ages; England; 2019/2020

| | Counts | Percentage of all emergency inpatient admissions |
|---|---|---|
| All inpatient admissions: in-year admissions by spell with any diagnosis codes | 17 183 220 | NA |
| Emergency inpatient admissions: in-year admissions by spell with any diagnosis codes | 6 506 970 | 100 |
| Emergency inpatient admissions with a mention of any neurological code (including dementia and stroke): in-year admissions by spell | 1 380 975 | 21.2 |
| Emergency inpatient admissions with neurological emergency codes: in-year admissions by spell with specific neurological diagnosis codes | 248 455 | 3.8 |
| Emergency inpatient admissions with neurological emergency code as primary reason for admission: in-year admissions by spell with specific neurological diagnosis codes | 72 485 | 1.1 |

Source: OHID Neurology Dementia Intelligence using Hospital Episode Statistics Admited Patient Care, NHS Digital.

were visualised in a range of chart types. Other than for the data in table 1 and the associated summary text, the paediatric population was excluded from further analyses. Chi-squared tests were used to determine if the distribution of admission by primary code versus secondary was independent of diagnosis and to assess the independence of diagnosis with involvement of a neurologist at admission, later in the spell or never. A Cramérs V was used to assess independence of admission with indices of deprivation; this was stratified by a range of diagnosis type. For the first two analyses, a code assigned to admission is the unit of analysis, for the third analysis, the unit of analysis is admission of an individual.

### Patient and public involvement

This was a retrospective empirical research study based on a secondary analysis of routinely collected anonymised clinical data. Patient and public involvement was not applicable.

### RESULTS
### Emergency inpatient admissions with any neurological codes or neurological emergency codes

Table 1 shows totals for inpatient activity for all ages by admission type for 2019/2020. Of a total of over 17 million inpatient admissions, 6.5 million were emergency admissions. Among these, 1.4 million (21%) were emergency inpatient admissions with a mention of any neurological code (including dementia and stroke). Emergency inpatient admissions with a specific neurological emergency code from the derived list (online supplemental appendix 1) represented 3.8% of all emergency inpatient admissions, and such a code was the primary reason for admission in 1.1% of all emergency inpatient admissions. Among all emergency adult in-year admissions by

spell with specific neurological diagnosis codes from the derived list, 132 485 (54.4%) comprised a single episode.

The number of unique adult patients (using HESID as a proxy) with in-year admissions by spell with specific neurological diagnosis codes from the derived list was 167 735; 75.6% (126 800) of these individuals have only one admission.

### NEUROLOGICAL CONDITIONS

Table 2 shows adult emergency inpatient hospital admissions with a mention of neurological emergency codes. Among a total of 237 755 such admissions, 65 060 (27.4%) had a neurological emergency code as a primary diagnostic code on the starting episode. Similar percentages were noted for acute paralysis (24.5%), functional disorders (29.8%), acute nerve root/cord syndrome (30.7%) and encephalitis (34.2%).

However, we see a strikingly different pattern for epilepsy, subarachnoid haemorrhage (SAH) and head injury (HI). The highest number of in-year admissions was for epilepsy (145 995), yet the percentage of those with epilepsy as the primary diagnostic code was only 10.9%. By comparison, the primary diagnostic codes for HI appeared in 73.1% of 29 235 admissions for this condition, and similarly for SAH, the primary diagnostic code appeared in 57.9% of 18 505 admissions. The p-value of the association of diagnosis with primary code is $<10^{-6}$.

### Proportion of admissions under a neurologist

Figure 1 shows the percentage of in-year adult emergency hospital admissions with mention of a neurological emergency code under the care of a consultant in neurology or neurosurgery. Overall, among 237 755 such admissions, 3230 (1.4%) were under the care of a consultant neurologist and 10 760 (4.5%) were

**Table 2** Emergency inpatient hospital admissions with a mention of neurological emergency codes; adults aged 18 years and over; England; 2019/2020

| | In-year admissions | In-year admissions with primary diagnosis on starting episode | Primary diagnosis as % of in-year admissions |
|---|---|---|---|
| Neurological emergencies | 237 755 | 65 060 | 27.4 |
| Emergency acute nerve root/cord syndrome | 41 215 | 12 655 | 30.7 |
| Emergency acute paralysis | 2600 | 635 | 24.5 |
| Emergency encephalitis | 5240 | 1790 | 34.2 |
| Emergency epilepsy | 145 995 | 15 945 | 10.9 |
| Emergency functional disorders | 6545 | 1950 | 29.8 |
| Emergency head injuries | 29 235 | 21 365 | 73.1 |
| Emergency subarachnoid haemorrhage | 18 505 | 10 720 | 57.9 |

Source: OHID Neurology Dementia Intelligence using Hospital Episode Statistics Admited Patient Care, NHS Digital.

under the care of a consultant neurosurgeon, with the majority of the admissions under a neurosurgeon having SAH or HI diagnostic codes (see online supplemental table 1). Only 1315 (0.9%) of admissions with mention of an epilepsy diagnostic code were under the care of a consultant neurologist. The p-value of the association of diagnosis with care under a neurologist either at admission/later in the spell or not at all is $<10^{-6}$.

### Influence of socioeconomic deprivation

Figure 2 shows a proxy for the number of people admitted (total, functional, epilepsy and SAH) by deciles of deprivation using the Indices of Multiple

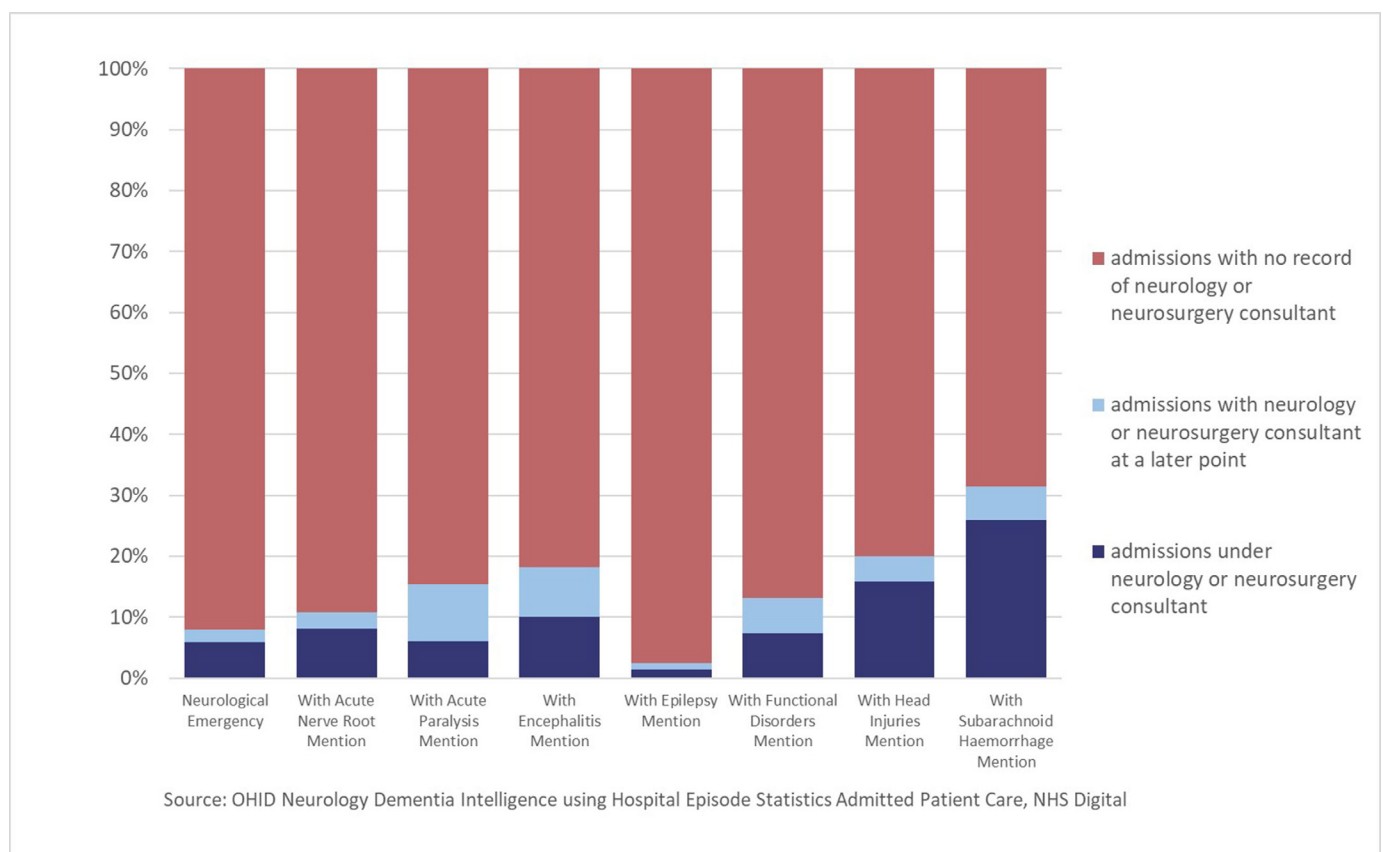

Source: OHID Neurology Dementia Intelligence using Hospital Episode Statistics Admitted Patient Care, NHS Digital

**Figure 1** Percentage of in-year emergency hospital admissions with mention of neurological emergency codes admitted under the care of a consultant neurologist or neurosurgeon; adults aged 18 years and over; England; 2019/2020; source: OHID neurology dementia intelligence using Hospital Episode Statistics admitted Patient Care, NHS Digital.

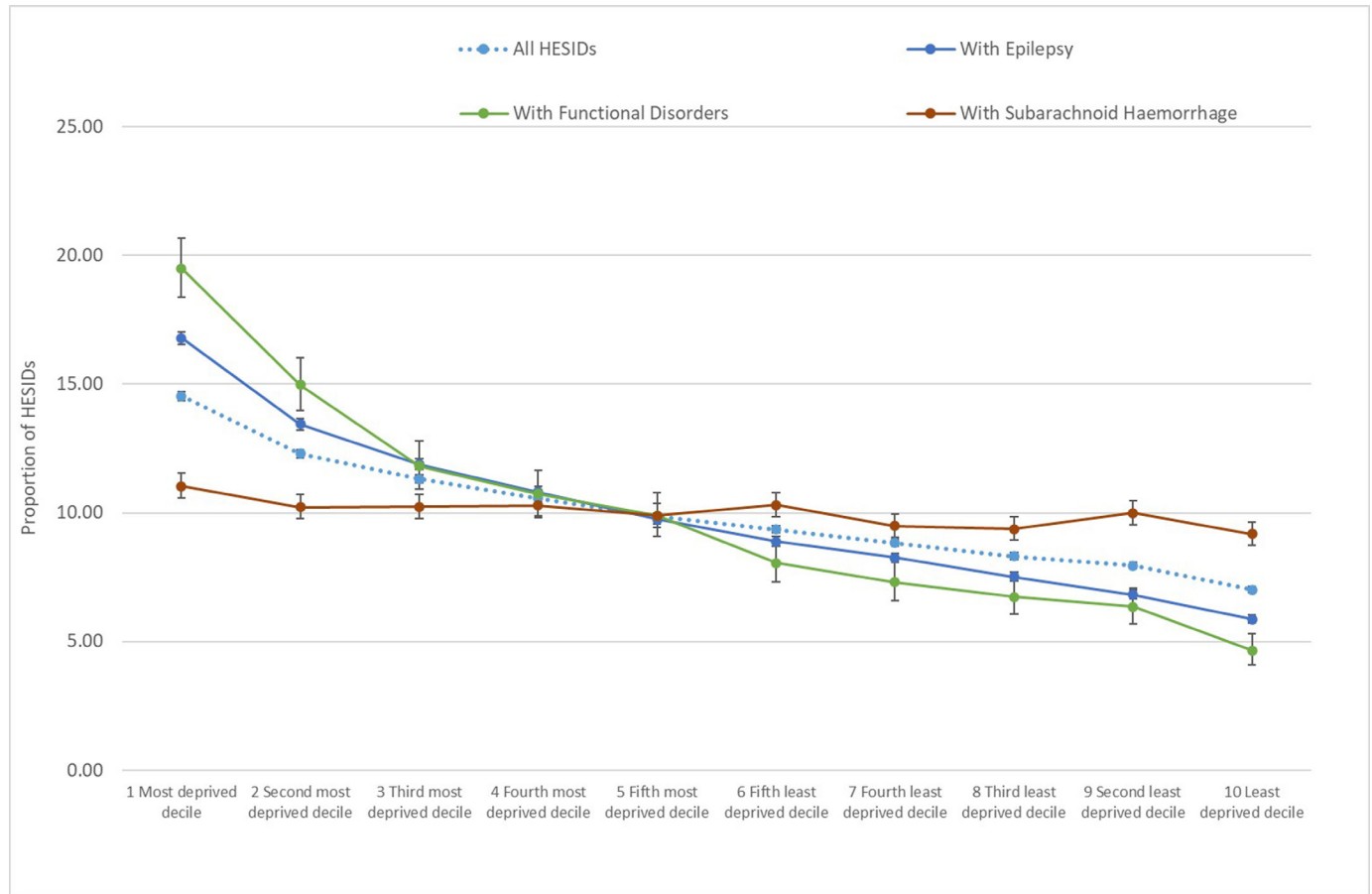

**Figure 2** Proportion of unique HESIDs (proxy for individuals) with an in-year neurological emergency hospital admission, by decile of deprivation and condition group; adults aged 18 years and over; indices of multiple deprivation 2019; England; 2019/2020; source: OHID neurology dementia intelligence using Hospital Episode Statistics admitted Patient Care, NHS Digital.

Deprivation[10] (see also online supplemental table 2). This shows that the number of people admitted varies significantly by social deprivation. This pattern is seen for the total admissions with mention of neurological emergency codes, but is more apparent for epilepsy and is most striking for FNDs, where the greatest number were in the most deprived and the fewest in the least deprived (p-values 1.6e-4, 2.5e-6 and 1.5e-8, respectively). By comparison, admissions with mention of SAH did not exhibit much variation by social deprivation (p-value=0.01).

## DISCUSSION

In this first large-scale analysis of 1.4 million emergency inpatient admissions in England for 2019/2020[8] with a mention of any neurological code, there were 248 455 with mention of a specific neurological emergency code from the derived list. The highest number was for epilepsy (145 995) with the next three most frequent causes being acute nerve root/spinal cord syndrome, HI and SAH; 1.4% of admissions with mention of a neurological emergency code were under a neurologist or neurosurgeon, with only 0.9% of those with mention of an epilepsy code under a neurologist. The greatest number of admissions

were in the most deprived, particularly for epilepsy and FNDs. Being able to determine the proportion of emergency admissions that constitute neurological emergency presentations is clearly crucial for service and clinical pathway design and workforce planning.

It was necessary to formulate a list of neurological emergencies and associated list of ICD-10 codes as a prelude to the analyses in this study. ICD-10 codes are designed and assigned with a focus on healthcare delivery rather than research. We took a pragmatic approach based on current UK practice, taking into account factors such as existing well-defined pathways of care (eg, acute stroke) or the absence of well-defined pathways (eg, FND). Although perhaps not classically considered neurological emergencies, FND presentations are increasingly important. This is with respect to their impact both on the individual and on healthcare resources, resources which are often used inappropriately and not in alignment with the needs of such patients. The approach taken to derive the list of neurological emergencies, as well as the approach to analysis of the data, took into account the nature of the data set being used.

We found that among adult emergency admissions with mention of a neurological emergency code, this was

the primary diagnostic code in 27.4% overall. However, despite epilepsy accounting for the greatest number of admissions in our data set, epilepsy appeared as the primary diagnostic code in only 10.9% by comparison with 73.1% for HI and 57.9% for SAH. It is important to note that epilepsy codes have been included among the neurological emergency codes but may reflect epilepsy as a comorbidity and not the reason for emergency admission, hence the comparatively low proportion of patients with epilepsy codes as the primary diagnostic code. A proportion of patients assigned epilepsy codes may have had dissociative seizures. Admissions with mention of epilepsy codes merit further analysis in future work, particularly given recent data on epilepsy-related mortality.[5] The current analysis cannot explain the differences between conditions in the proportion of relevant primary diagnostic codes being assigned. However, it is conceivable that the high proportion of HI admissions with HI as the primary diagnostic code, and similarly for SAH, may reflect the relative ease and pace of establishing the primary reason for admission.

We found that among emergency admissions with mention of a neurological emergency code, 1.4% were under the care of a consultant neurologist and 4.5% were under the care of a consultant neurosurgeon; only 0.9% with mention of an epilepsy diagnostic code were under the care of a consultant neurologist. Many admissions with single episodes not under neurology or neurosurgery also suggest many patients may not receive input from these disciplines despite having neurological codes assigned. Although the percentage of admissions under a consultant neurologist may appear disproportionately low given the greater number of neurologists by comparison with neurosurgeons, this may partly reflect the different balance of commitments between outpatient and inpatient pathways between neurology and neurosurgery. These findings are in keeping with previous data suggesting that a low proportion of epilepsy emergency admissions are under a neurologist. Further work, such as the prospective capture of neurological emergency codes with greater clinical engagement to improve their accuracy, is needed, but our findings are in general agreement with the recognised urgent need to address workforce capacity to support neurological emergency admissions.

The present study highlights additional concerns, particularly the apparent relationship between admissions for FND and socioeconomic deprivation. We know that in many areas in England, there is a significant gap in terms of provision of appropriate services for patients with FND,[11] very likely to increase the likelihood of re-presentation as an emergency. A better understanding of emergency admissions for FND is needed to determine how clinical pathways can be improved for such patients. The retrospective nature of the present study limits interpretation of findings such as acute nerve root/spinal cord syndrome being arguably more frequent than expected; one possibility is a rise in 'scan-negative' cauda equina syndrome, itself another potential FND.[9]

An important limitation of the methodology employed by this study is the inclusion of emergency admissions which mention neurological conditions, that is, admissions which include codes deemed, as part of this study, relevant for neurological emergencies. These codes could appear at any position in the patient's diagnostic coding, and during any episode(s) comprising the given spell; implicit in this approach is the possibility that the neurological condition was not the reason for the emergency admission. This, of course, is different to the inclusion of neurological emergencies per se, where the primary cause of the emergency admission was neurological. Previous work has raised concern about the accuracy of current data collection methods, including the erroneous assignment of neurological codes.[12]

## Conclusion

We have for the first time quantified the number of neurological emergency admissions across England, the distribution of conditions, the proportion under the care of a neurologist or neurosurgeon and evidence of variation by social deprivation. Further work, including improved clinical engagement with coding to ensure data accuracy, is required to understand the drivers of these findings. Further analyses are also needed to investigate issues such as the effect of hospital setting (eg, district general hospital vs neurosciences centre), the point in the spell at which the neurological diagnosis is made and the influence of neurologist involvement. This includes neurological emergency admissions for FND. These activities will inform clinical pathway change across primary and secondary care to reduce avoidable admissions and improve care.

**Author affiliations**
[1]Office for Health Improvement and Disparities (OHID), United Kingdom Department of Health and Social Care, UK
[2]Lancaster Medical School, Lancaster University, Lancaster, UK
[3]Department of Neurology, Lancashire Teaching Hospitals NHS Foundation Trust, Preston, UK
[4]Dementia Research Centre, National Hospital for Neurology and Neurosurgery, University College London Hospitals NHS Foundation Trust, London, UK
[5]Neurology Department, Great Western Hospital NHS Foundation Trust, Swindon, UK
[6]University Hospitals Birmingham NHS Foundation Trust, Birmingham, UK
[7]Neuroscience Clinical Reference Group, NHS England, London, UK

**Contributors** MJ, MS, JK and HE conceived and designed the work and jointly undertook the analysis and interpretation of the data. MJ, JK and HE jointly drafted the manuscript and revised it critically for important intellectual content. AW contributed to the conception of the work. JW, MM, CM, PM and AW contributed to the development of the derived list of neurological emergency diagnostic codes. All authors approved the final version. The corresponding author attests that all listed authors meet authorship criteria and that no others meeting the criteria have been omitted. MJ (data and interpretation) and HE (interpretation and narrative) are joint guarantors of this work.

**Funding** This work was in part supported by Health Data Research UK, which is funded by the UK Medical Research Council, Engineering and Physical Sciences Research Council, Economic and Social Research Council, Department of Health and Social Care (England), Chief Scientist Office of the Scottish Government Health and Social Care Directorates, Health and Social Care Research and Development Division (Welsh Government), Public Health Agency (Northern Ireland), British Heart Foundation and the Wellcome Trust.

**Competing interests** None declared.

**Patient and public involvement** Patients and/or the public were not involved in the design, or conduct, or reporting or dissemination plans of this research.

**Patient consent for publication** Not required.

**Ethics approval** Not applicable.

**Provenance and peer review** Not commissioned; externally peer reviewed.

**Data availability statement** Data may be obtained from a third party and are not publicly available. Information relating to access to access to HES data used for this study can be found at https://digital.nhs.uk/data-and-information/data-tools-and-services/data-services/hospital-episode-statistics/users-uses-and-access-to-hospital-episode-statistics.

**ORCID iDs**
Joanne Knight http://orcid.org/0000-0002-7148-1660
Hedley C A Emsley http://orcid.org/0000-0003-0129-4488

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
