## [Reviewer comments · BMJ Open]

ARTICLE DETAILS

TITLE (PROVISIONAL)	Numbers and types of neurological emergencies in England and the influence of socio-economic deprivation: a retrospective analysis of Hospital Episode Statistics data
AUTHORS	Jackson, Michael; Szczepaniak, Marta; Wall, Jasmine; Maskery, Mark; Mummery, Catherine; Morrish, Paul; Williams, Adrian; Knight, Joanne; Emsley, Hedley

VERSION 1 – REVIEW

REVIEWER	Văcăraș, Vitalie Iuliu Hagieganu University of Medicine and Pharmacy Faculty of Medicine, Neurology Department
REVIEW RETURNED	02-Apr-2022

GENERAL COMMENTS	Dear Authors, First of all, I want to congratulate you for the innovative and distinctive paper elaborated and registered, along with the immersion in an otherwise unapproached research topic as the one that was addressed. I appreciate the given opportunity of reviewing the provided manuscript. Comments and Suggestions After comprehensive considerations, I consider the material to possess valuable information regarding the management of neurological emergencies admission, picturing a clear representation of some the most sensible aspects that would not be commonly observable for a practitioner without a specific intervention. Nevertheless, I believe that several aspects should be reconsidered / explained in a more detailed manner. First of all, the results section comprises both the total number of emergency inpatient admissions with neurological emergency codes (without stroke or dementia - 248,455) and the adult emergency inpatient hospital admissions with neurological emergency codes (n = 237,755, without stroke or dementia). By comparison, the pediatric population, which would be considered as the difference between the two groups, is not referred to in any aspect, given the fact that this category of patients was not previously excluded from the study. This should be stated more clearly in the text. While the term "epilepsy" can be found several times in the manuscript referred to as a an emergency with a neurological
---

	code as primary reason for admission, a more suitable way of designating it should be as "status epilepticus" - In the way it was applied in the ICD10 list of codes. All the limitations of the current study should firstly be enumerated. Afterwards. each of them should be individually addressed.
--	---

REVIEWER	Cruz-Flores, Salvador Texas Tech University Health Sciences Center El Paso Paul L Foster School of Medicine, Department of Neurology
REVIEW RETURNED	19-Apr-2022

GENERAL COMMENTS	The authors present a descriptive study of the current burden of admissions to the hospital with a neurological emergency in the UK They found 1.4 millions admissions (21.1% of all admissions) corresponded to neurological emergencies More importantly they found that those with more social/financial deprivation had a higher burden Of note is that a priori the authors decided what specific neurological emergencies they wanted to count and as such I think they really limited the scope of their investigation for instance, they decided to exclude stroke although included subarachnoid hemorrhage the reason is not entirely clear for somebody outside the UK. Some of the codes are actually rather vague (ie acute paralysis) which can imply nerve, muscle, spinal cord or brain . I think the study would have been more useful having distinct pathologies, since the study was retrospective I am sure they should have been able to obtain discharge diagnoses No particular reason is given to exclude for instance Coma or meningitis. Status epilepticus seems included within epilepsy. On one hand the authors have large vague groups and on the other have specific groups like subarachnoid hemorrhage How many of the emergencies are traumatic vs medical/neurological The information of neurologist and neurosurgeon involvement is important as it shows problems with access to care I suspect the study may be underestimating the number of people presenting to the emergency room and being admitted with a neurological emergency The 2 most important points I can glean from this is neurological burden increase with deprivation and lack of access to neurological care The discussion is hard to follow it seems to be lacking cogency
---

REVIEWER	Jia, Jianping Capital Medical University, Innovation Center for Neurological Disorders and Department of Neurology
REVIEW RETURNED	27-Apr-2022

GENERAL COMMENTS	This study is the first large-scale analysis of neurological emergency admissions in England, the authors determined the number and types of neurological emergencies, and the influence
--

	of social-economic deprivation. However, there are still some points I'm confused.  1. "The highest number of in-year admissions for adults was for epilepsy (145,995), with epilepsy as the primary diagnostic code in 10.9% of these." (page3, line21-22). Does the "in-year admissions for adults" refer to "emergency inpatient admissions with a mention of any neurological code" or "with mention of a specific neurological emergency code"? 2. It is recommended to modify all the tables to three-line tables. 3. The Figure 2, which only shows the variation trend of the proportions of different diseases in people with different levels of social deprivation, can be further modified to indicate whether the differences among different populations are statistically significant. 4. The results of this study reflect the situation in 2019-2020, which coincided with the global COVID-19 pandemic. This study can further explore whether COVID-19 has an impact on the results, compared to the data before the outbreak of COVID-19. 5. This study could analyze whether a consultant neurologist or neurosurgeon have an impact on the prognosis of different diseases. 6. Social deprivation includes many factors, and it can be further discussed which factor has the most impact on admission. 7. In the results section (page10, line7-10), it was mentioned that the admission for epilepsy and functional neurological disorders was most affected by social deprivation, while SAH was on the contrary. However, the underlying reasons were not analyzed in the discussion section.
--	---

VERSION 1 – AUTHOR RESPONSE

Reviewer: 1

First of all, I want to congratulate you for the innovative and distinctive paper elaborated and registered, along with the immersion in an otherwise unapproached research topic as the one that was addressed. I appreciate the given opportunity of reviewing the provided manuscript. After comprehensive considerations, I consider the material to possess valuable information regarding the management of neurological emergencies admission, picturing a clear representation of some the most sensible aspects that would not be commonly observable for a practitioner without a specific intervention.

Response: We are grateful to the review for their very positive comments

Nevertheless, I believe that several aspects should be reconsidered / explained in a more detailed manner.

First of all, the results section comprises both the **total number** of emergency inpatient admissions with neurological emergency codes (without stroke or dementia - 248,455) and the **adult** emergency inpatient hospital admissions with neurological emergency codes (n = 237,755, without stroke or dementia). By comparison, the pediatric population, which would be considered as the difference between the two groups, is not referred to in any aspect, given the fact that this category of patients was not previously excluded from the study. This should be stated more clearly in the text.

Response: We thank the reviewer for this comment to permit clarification. Although table 1 does indeed include data on admissions across all ages, this is provided for context. The paediatric population was not included in further analyses. We have included a comment in the methods section to reflect this. "Other than for the data in table 1 and the associated summary text, the paediatric population was excluded from further analyses".

While the term "epilepsy" can be found several times in the manuscript referred to as an emergency with a neurological code as primary reason for admission, a more suitable way of designating it should be as "status epilepticus" - In the way it was applied in the ICD10 list of codes.

Response: The ICD10 code list used did include G409 'epilepsy unspecified'. This is because we intentionally included emergency admissions with mention of epilepsy as not all emergency admissions relating to epilepsy are due to status epilepticus.

All the limitations of the current study should firstly be enumerated. Afterwards, each of them should be individually addressed.

Response: A strengths and limitations section has been added, as per the editor's instructions, together with further text in the discussion section.

Reviewer: 2

The authors present a descriptive study of the current burden of admissions to the hospital with a neurological emergency in the UK

They found 1.4 million admissions (21.1% of all admissions) corresponded to neurological emergencies

More importantly they found that those with more social/financial deprivation had a higher burden

Response: We thank the reviewer for their insightful comments on the important findings on this study

Of note is that a priori the authors decided what specific neurological emergencies they wanted to count and as such I think they really limited the scope of their investigation

for instance, they decided to exclude stroke although included subarachnoid hemorrhage the reason is not entirely clear for somebody outside the UK.

Response: The existing methods section does include this statement "Of particular note, stroke, despite being a classical neurological emergency, was excluded from this exercise given that stroke is already the subject of the existing national audit system (SSNAP)." We hope this clarifies why stroke was excluded.

Some of the codes are actually rather vague (ie acute paralysis) which can imply nerve, muscle, spinal cord or brain.

I think the study would have been more useful having distinct pathologies, since the study was retrospective I am sure they should have been able to obtain discharge diagnoses

No particular reason is given to exclude for instance Coma or meningitis. Status epilepticus seems included within epilepsy.

On one hand the authors have large vague groups and on the other have specific groups like subarachnoid hemorrhage

Response: As the reviewer has highlighted, we did indeed include distinct pathologies such as subarachnoid haemorrhage. Discharge diagnoses data have been included by virtue of the assignment of ICD10 codes utilising this information.

How many of the emergencies are traumatic vs medical/neurological

Response: We have included a limitations section in accordance with the editor's instructions that covers this point

The information of neurologist and neurosurgeon involvement is important as it shows problems with access to care

Response: We agree

I suspect the study may be underestimating the number of people presenting to the emergency room and being admitted with a neurological emergency

Response: We agree; this study represents the largest analysis of its type and highlights the need for further research in this area

The 2 most important points I can glean from this is neurological burden increase with deprivation and lack of access to neurological care

Response: As mentioned above, we agree

The discussion is hard to follow it seems to be lacking cogency

Response: The discussion has been edited in response to reviewers' and editor's comments

Reviewer: 3

This study is the first large-scale analysis of neurological emergency admissions in England, the authors determined the number and types of neurological emergencies, and the influence of social-economic deprivation. However, there are still some points I'm confused.

1. "The highest number of in-year admissions for adults was for epilepsy (145,995), with epilepsy as the primary diagnostic code in 10.9% of these." (page3, line21-22). Does the "in-year admissions for adults" refer to "emergency inpatient admissions with a mention of any neurological code" or "with mention of a specific neurological emergency code"?

Response: This refers to the latter

2. It is recommended to modify all the tables to three-line tables.

Response: We would like to defer to BMJ Open formatting requirements on this point. We have had no editorial instruction to modify the tables.

3. The Figure 2, which only shows the variation trend of the proportions of different diseases in people with different levels of social deprivation, can be further modified to indicate whether the differences among different populations are statistically significant.

Response: Confidence intervals are included on the plot and a test of significance for distribution of admission by deprivation decile has been included and a Cramér's V test for correlation has been undertaken.

4. The results of this study reflect the situation in 2019-2020, which coincided with the global COVID-19 pandemic. This study can further explore whether COVID-19 has an impact on the results, compared to the data before the outbreak of COVID-19.

Response: This study related almost entirely to the pre-pandemic period. This is because the year '2019-2020' refers to the financial year ending at the start of April 2020. Thus, we would expect only negligible effects from the COVID-19 pandemic.

5. This study could analyze whether a consultant neurologist or neurosurgeon have an impact on the prognosis of different diseases.

Response: We thank the reviewer for the suggestion but this analysis is not feasible – patients with particular conditions would have been under the care of either a neurologist or a neurosurgeon, where an admission was under the care of a neurosciences clinician – eg patients with SAH under neurosurgery or patients with epilepsy under neurology

6. Social deprivation includes many factors, and it can be further discussed which factor has the most impact on admission.

Response: We entirely agree, but we only had access to data on the index of multiple deprivation. Certainly, future work could look at other relevant aspects such as lifestyle if data are available.

7. In the results section (page10, line7-10), it was mentioned that the admission for epilepsy and

functional neurological disorders was most affected by social deprivation, while SAH was on the contrary. However, the underlying reasons were not analyzed in the discussion section.

Response: We agree that this is extremely interesting, but to comment further on this would be to risk over-interpretation and take us into the territory of speculation. Further research is undoubtedly needed in this area.

VERSION 2 – REVIEW

REVIEWER	Cruz-Flores, Salvador Texas Tech University Health Sciences Center El Paso Paul L Foster School of Medicine, Department of Neurology
REVIEW RETURNED	15-Aug-2022

GENERAL COMMENTS	The study has limitations as does not seem to include syndromes or diagnosis such as coma. However it does establish at least 3 things: 1 neurological emergencies are common even when stroke is not included in the analysis; 2. patients with neurological emergencies rarely are under the care of a neurologist and ;3. social deprivation increases the probability of not having a neurological evaluation. It is clear more research is needed in this area such as neurological work force needed to cope with need, effect and interventions to improve access among socially deprived etc
--

REVIEWER	Jia, Jianping Capital Medical University, Innovation Center for Neurological Disorders and Department of Neurology
REVIEW RETURNED	11-Aug-2022

GENERAL COMMENTS	This is a very meaningful study and this field tends to be overlooked. It is detailed to clarify the reasons for the exclusion of stroke and the inclusion of functional neurological disorders (FNDs), however, the criteria of 'derived list' is vague and it is better to specify what factors have been considered when refined the list. In addition, FND has been included in the derived list considering the inappropriate use of resources, and it is needed for the improvement of the clinical pathways for such patients. Could you talk about what further work is needed based on your findings to achieve this aim?
--

VERSION 2 – AUTHOR RESPONSE

Reviewer: 3

Dr. Jianping Jia, Capital Medical University

Comments to the Author:

This is a very meaningful study and this field tends to be overlooked. It is detailed to clarify the reasons for the exclusion of stroke and the inclusion of functional neurological disorders (FNDs), however, the criteria of 'derived list' is vague and it is better to specify what factors have been considered when refined the list.

Response: We have amended the text in the methods section to clarify that this exercise was undertaken with reference to entities classically considered to be neurological emergencies. We have

also added some further text to explain that this work was based on a dataset generated for healthcare delivery rather than research, and that by necessity a pragmatic approach had to be taken to the derived list. We trust this addresses the reviewer's comment in this regard.

In addition, FND has been included in the derived list considering the inappropriate use of resources, and it is needed for the improvement of the clinical pathways for such patients. Could you talk about what further work is needed based on your findings to achieve this aim?

Response: We agree, and we have amended the text to make it clear that further analyses, needed to investigate issues such as the effect of hospital setting (eg district general hospital vs neurosciences centre), the point in the spell at which the neurological diagnosis is made and the influence of neurologist involvement, also applies to neurological emergency admissions for FND.

Reviewer: 2

Prof. Salvador Cruz-Flores, Texas Tech University Health Sciences Center El Paso Paul L Foster School of Medicine

Comments to the Author:

The study has limitations as does not seem to include syndromes or diagnosis such as coma. However it does establish at least 3 things: 1 neurological emergencies are common even when stroke is not included in the analysis; 2. patients with neurological emergencies rarely are under the care of a neurologist and ;3. social deprivation increases the probability of not having a neurological evaluation. It is clear more research is needed in this area such as neurological work force needed to cope with need, effect and interventions to improve access among socially deprived etc

Response: We agree, and we have covered these points in the manuscript.

We hope that you now find the manuscript suitable for acceptance but please do let us know if there are any remaining concerns.

Yours sincerely

Prof Emsley, on behalf of all co-authors